# Exploring the Lifetime Effect of Children on Wellbeing Using Two-Sample Mendelian Randomisation

**DOI:** 10.3390/genes14030716

**Published:** 2023-03-14

**Authors:** Benjamin Woolf, Hannah M. Sallis, Marcus R. Munafò

**Affiliations:** 1School of Psychological Science, University of Bristol, Bristol BS8 1TU, UK; 2MRC Integrative Epidemiology Unit, University of Bristol, Bristol BS8 2BN, UK; 3MRC Biostatistics Unit, University of Cambridge, Cambridge CB2 1TN, UK; 4Centre for Academic Mental Health, Population Health Sciences, Bristol Medical School, University of Bristol, Bristol BS8 2BN, UK

**Keywords:** Mendelian randomisation, wellbeing, family size

## Abstract

Background: Observational research implies a negative effect of having children on wellbeing. Objectives: To provide Mendelian randomisation evidence of the effect of having children on parental wellbeing. Design: Two-sample Mendelian randomisation. Setting: Non-clinical European ancestry participants. Participants: We used the UK Biobank (460,654 male and female European ancestry participants) as a source of genotype-exposure associations, the Social Science Genetics Consortia (SSGAC) (298,420 male and female European ancestry participants), and the Within-Family Consortia (effective sample of 22,656 male and female European ancestry participants) as sources of genotype-outcome associations. Interventions: The lifetime effect of an increase in the genetic liability to having children. Primary and secondary outcome measures: The primary analysis was an inverse variance weighed analysis of subjective wellbeing measured in the 2016 SSGAC Genome Wide Association Study (GWAS). Secondary outcomes included pleiotropy robust estimators applied in the SSGAC and an analysis using the Within-Family consortia GWAS. Results: We did not find strong evidence of a negative (standard deviation) change in wellbeing (β = 0.153 (95% CI: −0.210 to 0.516) per child parented. Secondary outcomes were generally slightly deflated (e.g., −0.049 [95% CI: −0.533 to 0.435] for the Within-Family Consortia and 0.090 [95% CI: −0.167 to 0.347] for weighted median), implying the presence of some residual confounding and pleiotropy. Conclusions: Contrary to the existing literature, our results are not compatible with a measurable negative effect of number of children on the average wellbeing of a parent over their life course. However, we were unable to explore non-linearities, interactions, or time-varying effects.

## 1. Introduction

A well-replicated [1,2,3,4,5,6,7,8] but contested [9,10,11,12,13,14] finding in the observational quantitative social science literature is a negative association between having children and subjective wellbeing in English-speaking countries. Although many studies have failed to replicate the finding in non-English-speaking countries [15,16,17], others have [18,19]. For example, Novoa and colleagues found a negative association between having children and subjective wellbeing in Chile [20]. Matters are further complicated by non-linearities depending on the age at which wellbeing is assessed in the parents. According to some of the studies, having children is not negatively associated with wellbeing when measured in geriatric populations, especially for those with a lower socioeconomic position (SEP), despite a negative association when measured in younger age groups [18,20,21,22,23]. This is possibly because of the increased social support that children can provide to their parents in old age [2,14,21,22].

Because these findings come from mostly cross-sectional studies, they should be treated with caution. For example, most of the studies cited above adjusted for only a few potential confounding variables (e.g., age, sex, and socio-economic position), if any, raising a question of residual confounding. Indeed, Deaton and Stone found that the choice of covariates adjusted for could produce radically different conclusions [15]. There will also likely be underlying psychological differences between adults who choose to have children and those who choose not to have children. Since the examined studies did not adjust for these variables, if these psychological differences also directly influence wellbeing, then they would be an additional source of unmeasured confounding. Relatedly, because most of the literature is derived from correlational surveys, these studies cannot ascertain the direction of effect [6]. Finally, cross-sectional studies cannot determine if the apparent change in effect with age is a cohort effect instead, although the corroboration of this finding by some prospective studies makes this less plausible [8]. As observational studies, these cohort studies are likely to still suffer from residual confounding.

Because of the higher risk of bias in traditional observational studies, it is becoming increasingly common in social epidemiology and econometrics to triangulate evidence from conventional observational studies with quasi-experimental designs [24]. One such design is Mendelian randomisation (MR) [25,26,27]. In a randomised controlled trial, participants are randomised to an intervention or control arm and followed up for a certain period. Because genetic variants are inherited at random, comparing the outcome status of an individual with and without a causal variant for the exposure is essentially analogous to a clinical trial [28]. In addition, because our genotype is fixed at conception, MR estimates are robust to reverse causation. This also means that any effect estimate derived from MR studies should be interpreted as the lifetime effect of the exposure. Applications of MR have traditionally focused on biomedical exposures where the analogy between MR and randomised controlled trials for a pharmacological intervention is strong because most drugs target proteins, which are the proximal product of genes [29].

MR has been gaining popularity as a method for answering causal questions in medicine, psychology and the social sciences in recent years [30,31]. The ‘social environment’, defined as the behaviour (and consequences) of those around us, is a potentially important cause of ill health and is a psychological mechanism. For example, one person’s smoking could cause those around them to smoke as well (‘social smoking’). Psychiatric genetics and evolutionary theory both suggest that the behaviours of those close by will associate with that person’s genotype. These are postulated to occur through gene–environment correlation, which may be active (when someone’s heritable phenotype influences someone to select into an environment) or evocative (when heritable phenotype evokes a change in the environment) and through the extended phenotype (when genes associate with the environment due to the effects of the heritable phenotype) [32,33]. For example, someone’s genetic predisposition to smoke may associate with their exposure to passive/environmental tobacco smoke, either by provoking social smoking or by homophily (the tendency of people to be friends with similar people). However, to the best of our knowledge, MR has not been used to explore the effects of social environmental exposures. This is, in part, because it is conceptually less clear as to how the potential effect of a genetic variant, which robustly increases the probability of a social environmental exposure—such as a traffic accident—by a very small amount, is genuinely equivalent to an exposure such as being hit by a lorry. Children, on the other hand, are a part of one’s social environment, but can plausibly be studied with a genetic design such as MR because having children is the primary endpoint through which evolution by natural selection occurs, and hence should be influenced by genetics [34,35]. However, the effects that children have on their parents after their birth is a social, rather than a direct biological, effect.

Therefore, exploring the effect that children have on their parent’s wellbeing not only answers a question of societal importance, but can also support the use of leveraging an active gene–environmental correlation within an MR study design to study the causal effect of environmental exposures. We therefore used a two-sample MR to explore the lifetime effect of having children on wellbeing.


**Strengths and limitations of this study**


Mendelian randomisation (MR) is a natural experiment that is theoretically robust for confounding and reverse causation.We were able to use two negative control analyses to explore the robustness of our study to two potential sources of residual confounding (populations structure and passive gene–environment correlation).We additionally used pleiotropy robust estimates (such as MR-PRESSO, MR-Egger, weighted median, and weighed mode) to explore if our result was affected by the direct effects of the genetic variants on the outcome not mediated by the exposure.Because we used summary data, we were unable to explore interactions or non-linear, time-varying, or time sensitive effects.

## 2. Methods

### 2.1. Study Design

We performed a two-sample MR analysis to explore the lifetime effect that children have on parental wellbeing. Specifically, we used the UK Biobank (UKB) as a source of genetic instruments [36,37], as well as their weights, for the number of children an individual has, and we used the 2016 Social Science Genetics Consortium (SSGAC) Genome Wide Association Study (GWAS) meta-analysis of subjective wellbeing as a source of instrument–outcome associations [38].

### 2.2. Data Sources

#### 2.2.1. UK Biobank (UKB)

The UKB is a large (~500,000 participants) population cohort study in the UK. Members of the public between the ages of 38 and 73 and who lived within 22 miles of an assessment centre were invited to participate from 2006 to 2010. Approximately 9.2 million individuals were invited to take part, with around 6% participating in the baseline assessment. The sample was 55% female, and predominantly of European ancestry (96%). The study design, participants and quality control (QC) methods have been described in full elsewhere [36]. UKB received ethics approval from the North West Multi-Centre Research Ethics Committee (REC reference 11/NW/0382). All participants provided written informed consent to participate in the study.

#### 2.2.2. Social Science Genetics Consortia (SSGAC)

Data on subjective wellbeing was extracted from the 2016 SSGAC subjective wellbeing GWAS (OpenGWAS ID: ieu-a-1009) [38]. This was a meta-analysis of 298,420 individuals from 59 studies. Samples included men and women, mostly of European descent, living in Europe, North America, or Australia.

#### 2.2.3. Within Family Consortium (WFC)

Data on wellbeing was also taken from the 2022 Within Family Consortium (WFC) GWAS (OpenGWAS ID: ieu-b-4851) [39]. This GWAS used the genetic overlap of relatives to adjust for which SNPs were inherited and therefore fully removed most plausible sources of confounding, including ancestry and genetic nurture [40]. The consortium combines data on almost 160,000 pairs of relatives from 17 cohorts. The GWAS itself had an effective sample size of 22,656 (male and female) individuals of European ancestry. A more detailed description of the individual cohorts included can be found in the paper’s Appendix A [39]. To avoid confusion, the WFC does not estimate between-relative effects, but instead the association between one person’s genotype and another person’s phenotype. The WFC uses genetic data from biological relatives to then remove biases in population genetics studies such as assortative mating and population structure.

### 2.3. Phenotyping

#### 2.3.1. UKB

Information on the number of (biological) children (OpenGWAS ID: ieu-b-4760) a participant had, number of full sisters (UKB ID: 1883, OpenGWAS ID: ukb-b-5593), number of full brothers (UKB ID: 1873, OpenGWAS ID: ukb-b-4263), general happiness (UKB ID: 20458, OpenGWAS ID: ukb-b-4062), number of older siblings (UKB ID: 5057, OpenGWAS ID: ukb-b-1997), and hair colour (UKB ID: 1747, Open GWAS IDs: ukb-d-1747_5, ukb-d-1747_4, ukb-d-1747_3, ukb-d-1747_1, ukb-d-1747_2, ukb-d-1747_6) were collected through a questionnaire asked either during the initial visit to an assessment centre or, in the case of general happiness, in an online follow up. Number of biological children was measured by asking men (UKB ID: 2405) “How many children have you fathered?”, and women (UKB ID: 2734) “How many children have you given birth to? (Please include live births only)”. The exact questions asked for the other phenotypes are provided in the Appendix A.

#### 2.3.2. SSGAC

This study included measures of life satisfaction, positive affect, or both in the GWAS. The specific questionnaires used for phenotype subjective wellbeing in each sample are described the Appendix A [38]. These were standardised for the meta-analysis. In most of the samples used in the SSGAC GWAS, a 1 to 2 standard deviation change is equivalent to a 1 unit increase on a 5-level psychometric question. For example, the 23andMe study asked participants to rate from very dissatisfied with their life (score = 0) to very satisfied with it (score = 5) and had a standard deviation of 1. This means that a one standard deviation increase would be the same as going from very dissatisfied to somewhat dissatisfied. Appendix A provide the gene-exposure and gene-outcome associations used in this study.

#### 2.3.3. WFC

All participating cohorts measured wellbeing using a questionnaire. Wellbeing measures were standardised prior to the meta-analysis. More details on phenotyping are provided in the Appendix A of the original paper [39].

### 2.4. Statistical Analysis

#### 2.4.1. Overview of the Analysis

The primary analysis was an inverse variance weighted meta-analysis of the Wald ratio using independent genome-wide significant (*p* < 5 × 10^−8^) SNPs for number of children identified in UKB as instruments [37]. The Wald ratios were defined as the variant’s association with wellbeing divided by the variant’s association with number of children.

We additionally used a number of sensitivity analyses, including (a) five pleiotropy robust estimators (MR-Egger, MR-RAPS, MR-PRESSO, weighted median, and weighted mode), (b) two sets of negative controls (hair colour, and number of parental siblings) as a falsification test for the presence of residual confounders of the instrument–outcome association, (c) the Within-Family Consortium (WFC) as a more robust (but less well powered) outcome GWAS, and (d) a less stringent *p*-value threshold (*p* < 5 × 10^−6^) for selecting SNPs to increase power. More details can be found in the Appendix A.

#### 2.4.2. Instrument Construction

Genetic instruments were selected using a statistical criterion of having a genome-wide significant association with the exposure (*p* < 5 × 10^−8^). We additionally clumped the variants using an r^2^ of 0.001 and KB of 10,000, thereby ensuring that the instruments were independent of each other. Genetic variants that have their gene–exposure association estimated in the same dataset used to select instruments can suffer from a bias called ‘Winner’s Curse’. This occurs because variants can meet the statistical criteria due to having a genuine association or because they, by chance, have an unusually large amount of noise. This results in variants appearing to have a larger association than they actually do. We therefore further filtered these variants using the False discovery rate Inverse Quantile Transformation (FIQT) winners curse correction developed by the SSGAC [41]. This uses an analogy between multiple testing and winners curse to apply an easy-to-implement correction to effect estimates.

We used the TwoSampleMR R package to harmonise the two GWASs. Palindromic SNPs were only excluded if their allele frequency could not be used to infer which strand was positive. In cases where SNPs were missing in the outcome dataset, we used TwoSampleMR to automatically impute LD proxy variants, using an r^2^ of 0.8 from the European subsample of the 1000 genomes project.

MR assumes that the causal pathways go from the genetic variant to the exposure to the outcome. However, when selecting genetic instruments using a data-driven method, as we did, it is possible to select some SNPs wherein the causal path is from the variant to the outcome to the exposure. When the exposure and outcome GWASs are of similar power, one should expect SNPs to explain a larger proportion of the variance in the more proximal GWAS than the more distal one. Steiger filtering, which we applied, uses this logic as a method of removing SNPs that are more proximal to the outcome than the exposure.

#### 2.4.3. Statistical Methods

The primary MR estimator in this study was the Wald ratio. This is defined by the variant–outcome association divided by the variant–exposure association. Because both the variant–exposure and variant–outcome associations were derived from linear models, MR analysis also assumed a linear model.

We then used six methods of meta-analysing for the MR estimates of each SNP: IVW, MR-Egger, weighted median, weighted mode, MR-RAPS, and MR-PRESSO. The IVW estimate will return the true effect if all the IV assumptions are valid. The other five ‘pleiotropy robust’ methods can return the true effect if some of the instruments are invalid, but they have reduced power. Specifically, the weighted mode assumes that the modal effect size is a valid estimate of the true effect size, while weighted median assumes that at least half of the SNPs are valid. IVW can be thought of as a regression of the variant–outcome association on the variant–exposure association, with the intercept fixed at zero [42]. MR-Egger extends this model to allow for a non-zero intercept. If we assume that the variant–exposure effect size is independent of the size of any bias (such as a pleiotropic effect), then the biasing pathways should impact the intercept but not the slope. This assumption is called the INSIDE assumption. Additionally, MR-Egger assumes that there is no measurement error in the exposure GWAS (called the NOME assumption) [43,44]. MR-RAPS assumes that pleiotropic SNPs are outliers (modelled by a random effects parameter with a mean of zero) and therefore down weights them in a random effect meta-analysis. We implemented MR-RAPS using a square error loss function, accounting for overdispersion (i.e., systematic pleiotropy) [45]. MR-RAPS is robust to both balanced pleiotropy in non-outlier SNPs and weak instrument bias. MR-PRESSO first runs a global test for pleiotropy, and then tests for outliers. It then excludes outliers before running a meta-analysis and testing for a difference between the MR estimates with and without the outliers. MR-PRESSO assumes INSIDE and that >50% of SNPs are valid instruments [46].

#### 2.4.4. Assumptions of the Analysis

Two-sample MR is an extension of MR to a summary data setting. MR is itself an extension of Instrumental Variables (IV) analysis for genetics. IV makes three assumptions: (1) relevance—that the variant is robustly associated with the exposure, (2) independence—that there are no variant–outcome confounders, (3) exclusion restriction—that the variant causes the outcome only through the exposure. For the point estimate to be interpretable, IV analysis additionally has to make an identification assumption. Here, we make the NO Simultaneous Heterogeneity (NOSH) assumption [47]. This assumption is valid if the causes of heterogeneity in the variant–exposure association are not the same as the causes in the variant–outcome associations, or if either of these associations are homogeneous. Finally, two-sample MR makes two additional assumptions: (1) That the GWASs come from homogeneous populations. This is required for the MR estimate to be meaningful. (2) That there is no sample overlap. The effect of the second assumption is to make weak instrument bias deflationary [48,49,50].

Because two-sample MR uses summary data from previously conducted GWASs, if these studies assume, as they typically do, a linear effect, then it is not possible to explore non-linearity using summary data. Traditional IV estimators, such as the Wald ratio used in the two-sample MR, can still provide a valid estimate of the average causal effect in the presence of non-linearities [51].

#### 2.4.5. Assessment of Assumptions

Weak instrument bias is inversely proportional to the F-statistic of the variant–exposure association, with a common threshold of ten being required. We therefore calculated the mean F-statistic for the variant–exposure association, as well as the F-statistic for each SNP. MR assumes that there are no confounders of the variant–outcome association.

Although the other assumptions made in an MR analysis are not provable, it is possible to run falsification tests on many of them. To ensure that population structure was controlled for, we checked that the GWASs either used BOLT-LMM, which adjusts the GWAS for the entire genetic relationship matrix [52], or adjusted for the principal components of the genetic relationship matrix. LD score regression (LDSC) can be used to explore the residual population structure in GWASs that do not use linear mixed models such as BOLT-LMM. Specifically, if the LDSC intercept is very different from one, this implies the presence of a residual population structure [53]. We also ran a set of sensitivity analyses, described below, as falsification tests for residual gene–outcome confounders.

Horizontal pleiotropy occurs when a genetic variant associates with two phenotypes for independent reasons. This can cause a violation of the exclusion restriction assumption. However, if horizontal pleiotropy is present, the exact pathway should be different for each variant. It has therefore been argued that, if it occurs, it should create heterogeneity in the MR estimates, and its presence can therefore be tested using a heterogeneity statistic. The presence of horizontal pleiotropy was also visually explored using a funnel plot.

The NOME assumption can be tested by checking that the I^2^ statistic for the gene-exposure association is greater than 90% [43]. Since this is a measure of the variability in the variant–exposure association, we will refer to it as the I^2^_GX_ statistic for clarity.

We checked sample overlap by checking which samples were reported as being included in each of the GWAS consortia in the respective publications. Two-sample MR studies generally validate the assumption that samples are drawn from the same population by checking that the studies are demographically similar [50]. However, in situations where their instrument–exposure or instrument–outcome association has been measured in both the exposure and outcome samples, it can be possible to validate this assumption quantitively. One would expect to find only chance differences in estimates drawn from the same population [54,55]. Therefore, on top of comparing demographic information and because information on happiness from the UKB was used in the SSGAC GWAS, we checked that the average-difference SNP estimates from the UKB compared to the SSGAC and WFC were approximately zero. Because the differences in the precision of the measure of wellbeing between the UKB and SSGAC could introduce heterogeneity into the estimation of the difference in SNP effects, we chose a random-effects meta-analysis as the primary estimator.

### 2.5. Sensitivity and Additional Analyses

#### 2.5.1. Negative Controls

We used two sets of negative control outcomes to explore two potential sources of confounding. Firstly, hair colour is known to vary by ancestry in the UK/European population, but *prima facie* should not have a direct causal relationship with the number of children we have [56]. It can therefore be used as a negative control outcome for population structure (see Figure 1 for a visual representation). Secondly, wellbeing may be affected by our developmental environment. However, the number of children we have is affected by our parental genotype (through inheritance). The parental genotype will also influence our developmental environment because of the effects that the parental genotype will have on the parental phenotype and therefore on the developmental environment (i.e., through genetic nurture, see Figure 2 for a visual representation). To test if genetic nurture is a residual confounder, we used the number of siblings as a negative control outcome. The number of siblings an individual has is caused by parental genotype but is unlikely to be caused by the number of children that the individual has. Therefore, the most likely explanation of an MR association with number of siblings is residual confounding due to genetic nurture. We ran the negative controls only using an IVW estimator because it is the most efficient estimator and thus the most likely to detect an association if there is one. In addition, we wanted an estimator which was *not* robust to pleiotropy because a pleiotropic association between the genetic instrument and the negative control outcomes (i.e., which is not mediated by number of children) is still evidence of an association between the instrument and a confounder of the instrument–outcome association, and hence a violation of the independence assumption.

#### 2.5.2. WFC GWAS

Family data has been proposed as a way of eliminating the risk of confounding in MR studies [40]. Potential violations of the independence assumption, such as population structure and genetic nurture, occur because the distribution of SNPs in a population GWAS is only approximately random. By conditioning on the parental genotype, however, the back door path used by these biases is blocked. Although these GWASs are less biased, the use of non-independent observations means that they need much larger samples than a population GWAS to achieve the same level of power. We therefore used the WFC GWAS of well-being as an additional sensitivity analysis to explore the robustness to potential confounders.

#### 2.5.3. Less stringent SNP Selection

Power in two-sample MR studies is a function of instrument strength, the precision of the outcome GWAS, and the number of instruments. Since we are limited to using pre-collected data, the outcome GWAS’s precision cannot be varied. However, a *p*-value threshold of 5 × 10^−6^ equates to an F-statistic of approximately 10 and should therefore not lead to weak instruments, but because it is 100-fold larger, it should increase the number of SNPs used in the analysis. We therefore also used SNPs with an indicative association (*p* < 5 × 10^−6^) with the exposure to explore how sensitive the primary analysis was to a potentially better-powered set of instruments.

#### 2.5.4. Leave-One-Out Analysis

We additionally used a ‘leave-one-out’ analysis and the MR-PRESSO outlier test, as part of our MR-PRESSO analysis, to explore if any of the variants were outliers and had a disproportionate effect on the overall IVW estimate. The ‘leave-one-out’ sensitivity analysis works by excluding each SNP in turn and running the IVW analysis without the excluded SNP. We then visually inspected the leave-one-out plot for indication that a single SNP had a disproportionate effect on the MR results. The MR-PRESSO outlier test explores whether the observed values are different from the expected values based on a regression model of the variant–outcome associations on the variant–exposure associations. Additional information, including details on genotyping, instrument construction, and MR estimators, can be found in the Appendix A. This manuscript conforms to the STROBE-MR guidelines [57].

#### 2.5.5. Bidirectional Analysis

In addition to the analyses planed in our protocol, we included a bidirectional analysis in order to explore the possibility that reverse causation might explain the observational results. Specifically, we selected variants associated with well-being in the SSGAC GWAS at a 5 × 10^−6^
*p*-value threshold, and we used the FIQT Winner’s Curse correction. After harmonizing the UKB number of children GWAS with these variants, we applied Steiger filtering and estimated the MR estimates using MR-RAPS with overdispersion and a Huber loss function. This estimator was chosen because it is reasonably robust to both weak instrument bias and pleiotropy.

## 3. Results

### 3.1. Descriptive Data

#### 3.1.1. Number of Participants and SNPs in Each Stage

The UKB exposure GWAS had information from over 460,000 participants on almost 10 million SNPs. The primary outcome analysis used information on six of these SNPs, which were significant genome-wide for number of children from 298,420 participants from the SSGAC. This was increased to 50 SNPs by using a 5 × 10^−6^
*p*-value threshold. The WFC outcome data used information on eight genome-wide significant SNPs with an effective sample size of 22,656 participants (Figure 3).

#### 3.1.2. Two-Sample MR Specific Assumptions

Both outcome samples had some overlap with the UKB. The SSGAC does not state how many UKB participants were included; however, around 157,000 UKB participants provided information on the measure of general happiness used by the SSGAC, which entails a maximum sample overlap of around 53% for the SSGAC and 34% for the UKB. The UKB also contributed around 4250 sibships to the WFC wellbeing GWAS, which equates to around 19% of the WFC sample and 2% of the UKB sample.

Because all GWASs were drawn from European populations of both males and females with some overlapping participants, it seems likely that the samples can all be treated as coming from the same population. We also found no evidence of a difference in SNP effect estimates for the SNP–outcome associations in the UKB and the WFC or the SSGAC consortium using either a 5 × 10^−6^
*p*-value or 5 × 10^−8^
*p*-value threshold, which further supports this conclusion (Appendix A) [58].

### 3.2. Main Results

The primary IVW estimate does not indicates strong evidence of a negative (standard deviation) change in wellbeing (β = 0.153 (95% CI: 0.210 to 0.516) per child parented (Figure 4 and Appendix A).

### 3.3. Assessment of Assumptions

#### 3.3.1. Weak Instrument Bias and NOME

For the primary analysis, the F-statistic was 49, and the I^2^_GX_ for the instrument–exposure association was 98%. These both imply that there would be around a 2% error in the MR estimates due to weak instrument bias. For the analysis using the WFC, the F-statistic was 44, and the I^2^_GX_ was 98%. For the analysis using a less stringent *p*-value, the F-statistic was 25, and the I^2^_GX_ was 96%.

#### 3.3.2. Heterogeneity and Exclusion Restriction Violations

The Cochrane Q statistic for the Wald ratios of the primary analysis was 24.59 (*p* < 0.001), and the I^2^_GX_ for the Wald ratio was 80%. Together with the asymmetric funnel plot (Appendix A) and the MR-PRESSO global test for outliers (*p* = 0.006), this implies the presence of some pleiotropic SNPs. However, the Egger intercept was −0.004 (SE = 0.017, *p* = 0.815). Similar results were found for the secondary analyses (Appendix A).

### 3.4. Sensitivity and Additional Analyses

#### 3.4.1. Pleiotropy Robust Estimators

The pleiotropy robust estimates were mostly similar to the IVW estimate, although generally slightly deflated (Figure 4 and Appendix A). The exception to this was the MR-Egger estimate, although the wide 95% confidence interval (which overlaps with the IVW estimate) implies that this could be due to a lack of precision. In addition, the MR-RAPS estimate was slightly inflated, probably because RAPS is robust to both moderate amounts of weak instrument bias and pleiotropy.

#### 3.4.2. Negative Controls

The negative control outcome analysis did not find any evidence of an association of the instruments with hair colour; however, there was evidence of an association with two out of three of the sibling questions (*p* < 0.001 for number of full brothers and *p* = 0.005 for number of full sisters), implying the possibility of some residual confounding (Appendix A).

#### 3.4.3. WFC Outcome

Consistent with the negative control analysis, the WFC secondary analysis showed deflated point estimates compared to when using the SSGAC outcome GWAS (Appendix A). For example, the IVW estimate was −0.049 (95% CI: −0.533 to 0.044).

#### 3.4.4. Less Stringent SNP Selection

The standard error of the IVW estimate when using a 5 × 10^−6^ threshold was more than three times smaller than when using the more traditional 5 × 10^−8^ threshold. Appendix A presents the results of this sensitivity analysis for all estimators. MR-RAPS in particular, as a weak instrument robust estimator, still did not find strong evidence against the null hypothesis (β = 0.082, SE = 0.054, *p* = 0.134).

#### 3.4.5. Leave-One-Out Analysis and MR-PRESSO Outlier Test

The MR-PRESSO outlier test for the primary analysis identified rs10270358 and rs72687493 as outliers. An exploratory search of Phenoscanner showed no phenotypes associated with rs72687493 but found that rs10270358 is associated with seeing a doctor for anxiety or depression, as well as chronic disability/infirmity, both of which could reduce wellbeing [59]. However, these SNPs did not seem to introduce a bias in the leave-one-out analysis (Appendix A), and the outlier test did not detect any outliers in the secondary analyses.

#### 3.4.6. Bidirectional MR

Our bidirectional analysis found a negative association between wellbeing and number of children (β = −0.069, 95% CI −0.010 to −0.128), implying that having lower wellbeing causes people to have more children.

## 4. Discussion

Contrary to the existing observational literature, our results do not imply that children have a detrimental effect on parental wellbeing. If we assume the measures of wellbeing in the SSGAC are on a ratio scale, and that people score the nearest category to what they feel, then our 95% confidence interval would be compatible with either no effect or as much as a one-unit increase on this scale for each child someone has (e.g., from neither satisfied nor dissatisfied to somewhat satisfied), but incompatible with a measurable negative change in subjective wellbeing.

However, our additional and sensitivity analyses imply that our primary point estimate may overestimate the true effect. Our negative control analysis found that our instrument was associated with the number of siblings, and that the point estimate was deflated when using the WFC outcome GWAS, implying the presence of residual confounding due to genetic nurture. Likewise, the heterogeneity statistics implied the presence of residual confounding, while most pleiotropy robust estimators were again deflated. It is therefore likely that the true effect, if existent, will be smaller than a measurable change (i.e., a one-unit increase or decrease in a 5-level psychometric question) in well-being for every child a parent has.

Our bidirectional analysis indicated that having lower wellbeing causes people to have more children. This implies that reverse causation could account for some of the observational findings of a negative association between wellbeing and number of children. However, because of the small effect size (1 standard deviation decrease in wellbeing resulting in 0.07 more children), this is unlikely to entirely explain the observed association.

Our study’s results appear to contradict a recent study by Giannelis and colleagues (2021) [60]. They used observational data from the UKB and two-sample MR (using the UKB and Psychiatric Genomics Consortium) to explore the effects of cohabitation and having children on depression. Their headline results appear to imply that children increase the risk of depression using IVW. Since depression plausibly has a negative effect on wellbeing, this finding is superficially contradictory to ours [61]. However, none of the directional pleiotropy robust estimators show deflated estimates, and they do not find evidence of an effect. Since they use a liberal *p*-value threshold to select SNPs, this is less likely to represent low power. Thus, as with our study, it is likely that IVW has overestimated the true effect.

### 4.1. Pre-Specified Interpretation

In the study protocol, we pre-specified how we would interpret the findings of our sensitivity analyses. Specifically:

#### 4.1.1. Pleiotropy

Because the indicators for the presence of pleiotropy (such as the I^2^_GX_ and Cochrane Q statistics for the Wald ratio, funnel plot and MR-RAPS) all indicated the presence of pleiotropy, and because the ‘pleiotropy robust’ estimators generally had deflated estimates compared to the IVW estimate, it seems likely that the IVW estimates were inflated by some residual pleiotropy.

#### 4.1.2. Residual Confounding

The association of the instruments with two out of three of the sibling negative controls, combined with the change in estimate from the WFC GWAS, implies that there was some inflation due to residual confounding from genetic nurture in the primary IVW estimate.

#### 4.1.3. Low Power

The number of SNPs increased almost nine-fold in this secondary analysis when compared to the primary one. This resulted in a three-fold decrease in the size of the standard error (0.185 to 0.053) for the IVW estimate, and implies that there is also a large amount of residual random error in the estimates. However, the point estimates in this analysis were generally deflated when compared to the primary analysis. This should be in part explained by the approximate halving of the F-statistic.

### 4.2. Generalisability

One possible explanation for the discrepancy between the observational and MR estimates, other than residual confounding in the observational studies, is that the target estimates are not directly comparable. If we assume that age does not modify the variant–exposure association, then MR estimates should be interpreted as the average effect of the exposure on an outcome over the entire lifetime up to recruitment (typically in their late 50s at recruitment for the UK Biobank, which is the single largest study in the SSGAC) [62]. This means that the transient effects of having children on wellbeing, such as the stress of looking after a new-born baby, may not be detectable in a typical MR design. Some studies found that having children was beneficial to parental wellbeing in old age [18,20,21,22,23]. One possible explanation for the discrepancy between our results and the existing literature’s results would therefore be that the transient negative effect is counterbalanced by the later positive ones—resulting in an average effect close to zero. Likewise, there could be effects in age groups older than those included in our study. Although methods for addressing time-varying exposures are currently being developed [63], there is still no consensus on how to best estimate transient effects within an MR framework. We are therefore unable to empirically explore this interpretation further here. MR still produces a valid test for the average lifetime effect of exposure to genetically predicted levels of the exposure [64]. We therefore believe that our results are a valid, if blunt, measure of the total longitudinal effect up to recruitment. However, further quasi-experimental studies should also be leveraged to explore short term effects. For example, the short-term effects after birth could in theory be studied using an interrupted time-series design.

Additionally, our estimates were all drawn from European samples. Because some of the existing literature had found different effects in predominantly non-European non-English speaking populations compared to those observed in English-speaking populations [15,16,17], our results may not generalise to other populations.

### 4.3. Strengths and Limitations

This study has several methodological strengths. Firstly, we believe it is the first study to apply MR to explore the effects of the heritable environment in a setting in which gene–environment equivalence is plausible. By doing so, we have been able to leverage the methodological strengths of MR, such as improved robustness to confounding and reverse causation to explore the effect of having children on wellbeing.

Secondly, this is the first applied study to have implemented the MRSamePopTest R package, a novel test of the two-sample MR ‘same-population’ assumption described in detail elsewhere [58]. We are unaware of other existing tests of this assumption and therefore hope that it will be useful in future applied MR studies.

There are also methodological limitations to this application of two-sample MR. As already noted, we were unable to explore time-sensitive effects. Relatedly, we were forced to assume a linear dose–response relationship for the effect of the number of children on wellbeing. We considered a sensitivity analysis using individual-level data, but we ultimately decided against doing so due to a lack of sufficiently good individual-level data; of the two available data sources, ALSPAC had detailed phenotyping, but on a relatively small number of participants (~2000). Because non-linear MR is less well powered than a linear MR, this sample would therefore be underpowered for this analysis. On the other hand, the UKB, only had a five-level minimal phenotype for happiness. Since poor phenotyping can mask non-linearities, and because happiness may not be the same as wellbeing, the interpretation of any analysis in the UKB would be unclear [65]. MR is generally more robust as a test of the causal null hypothesis than as a method of effect estimation because of many of the complications (such as those described above) of interpreting MR effect estimates. However, this approach to interpreting MR results may be less robust here because our sensitivity analyses implied that our study may be underpowered.

As with many MR studies using genome-wide significant SNPs as instruments, we have not explored the causal mechanism linking the SNPs to having children. However, provided the other Instrumental Variables’ assumptions are valid, IV analyses do not require a *causal* association between the instrument and the exposure [66]. Although there was substantive sample overlap between our exposure and the outcome GWAS, we do not expect there to be material bias to our MR results. The one-sample MR two-stage least square estimator is asymptotically equivalent to the two-sample MR Wald ratio, but it is necessarily applied in settings equivalent to 100% sample overlap. If two-sample MR is necessarily biased by sample overlap, then a one-sample MR would not provide valid estimates. Indeed, existing research implies that most two-sample MR estimators perform well even with a 100% sample overlap [67]. Instead, the primary effect of no sample overlap is to ensure that weak instrument bias attenuates results. However, the F-statistic in our primary analysis is 49, which implies only a 2% bias due to weak instruments. While we cannot guarantee this will attenuate results, it does not seem likely to cause material bias. Finally, we cannot explore more detailed psychological mechanisms, such as differences between expected (numbers of) children and reality, which could explain the observational analyses.

## 5. Conclusions

We conducted a two-sample Mendelian randomisation study to explore the causal effect that children have on parental wellbeing. Contrary to the previous literature, our results do not imply the presence of a negative lifetime effect of having children on wellbeing. Comparing our results to many existing observational studies is complicated by the temporal insensitivity of MR estimates. Future studies could therefore consider using other quasi-experimental methods, such as Interrupted Time Series [68], to explore if the discrepancy between our findings and the observational literature are due to the transient effects of having children, of which MR is either unable to detect or which average out over the life course.

## Figures and Tables

**Figure 1 genes-14-00716-f001:**
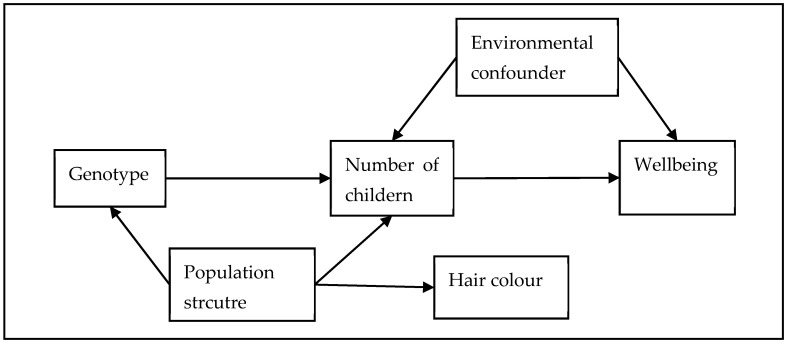
Directed acyclic graph for the hair colour negative control outcome. Hair colour is hypothesised to be associated with population structure, but to not have a direct causal link to either the genetic instruments or wellbeing. Therefore, any association between the genetic instruments and hair colour will be due to a residual confounding effect of population structure.

**Figure 2 genes-14-00716-f002:**
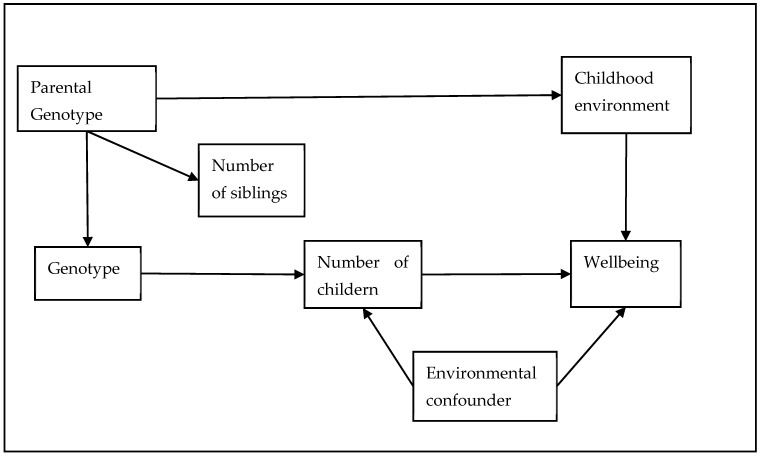
Directed acyclic graph for the number of siblings negative control outcome. The parental genotype determines not only their children’s genotype, but also, via the parental phenotype, the environment in which the children grew up. If someone’s childhood environment influences their wellbeing later in life, then the parental genotype would confound any association between an individual’s genetic instruments and wellbeing. Because, in the UK, parents typically stop having new children before their children start having children, it unlikely that a child having children will influence his or her parents to have more children. Hence, the number of siblings an individual has should not be caused by an individual’s genetic liability to having children, but will be influenced by the parental liability to having children. Therefore, any association between an individual’s genetic liability to having children and the number of siblings they have would be an indicator of residual confounding due to the parental genotype.

**Figure 3 genes-14-00716-f003:**
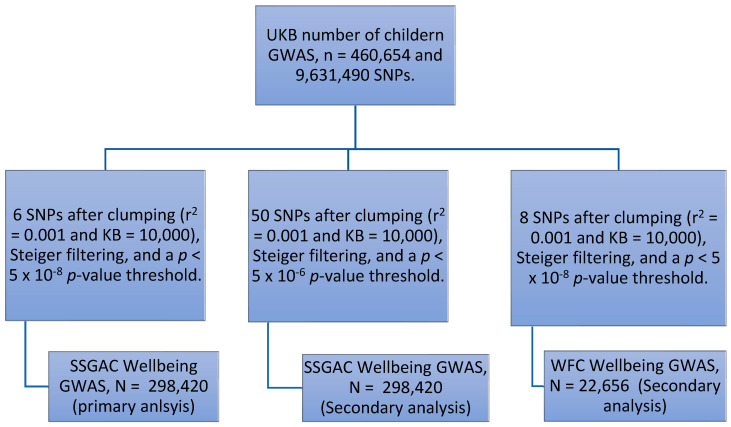
Flow chart of SNPs and participants.

**Figure 4 genes-14-00716-f004:**
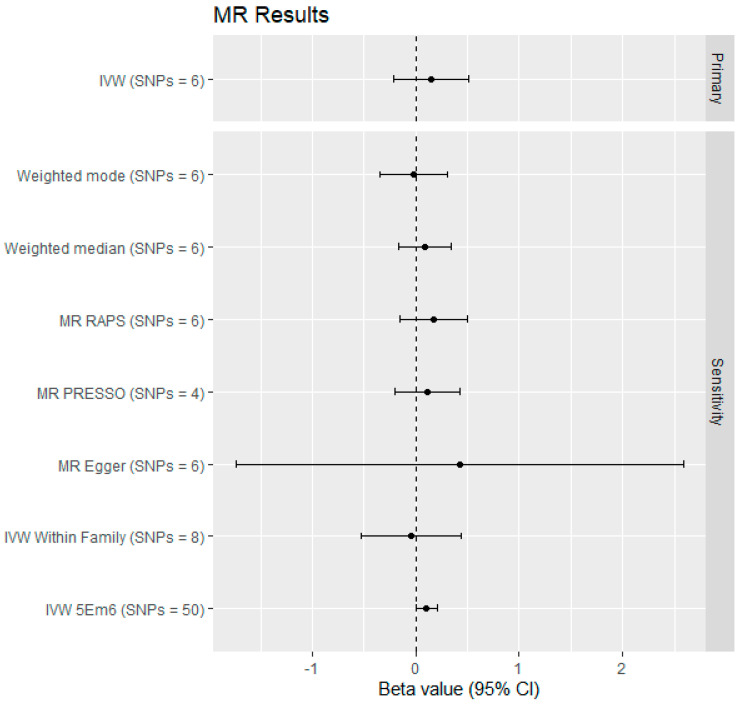
Forest plot of the primary and sensitivity analyses.

## Data Availability

All data that were used in this study are publicly available from the MRC-IEU OpenGWAS platform from the following links: UKB number of children—https://gwas.mrcieu.ac.uk/datasets/ieu-b-4760/ (accessed on 1 January 2023), SSGAC wellbeing—https://gwas.mrcieu.ac.uk/datasets/ieu-a-1009/ (accessed on 1 January 2023), and WFC wellbeing—https://gwas.mrcieu.ac.uk/datasets/ieu-b-4851/ (accessed on 1 January 2023). The R code used in the manuscript is available from https://doi.org/10.17605/OSF.IO/BTPH9.

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
