# Peer review of "Exploring the Lifetime Effect of Children on Wellbeing Using Two-Sample Mendelian Randomisation"

_genes, 2023, doi:10.3390/genes14030716_

Round 1

Reviewer 1 Report

Review of genes-2202080

Thank you for the invitation to review the manuscript “Exploring the lifetime effect of children on wellbeing using 2 two-sample Mendelian randomisation”.

I have included several comments and suggestion below and hope that these are helpful to the authors when revising their work.

Abstract

L22: revise “an inverse variance weighted analyses” to maintain consistency of singular vs. plural usage.

L23, 26: acronyms such as “GWAS” and “WFC” should be defined on first use.

Although the authors write “we estimate a … increase”, the confidence interval includes a possible null effect and indeed also negative effects of children on well-being. Could the wording be revised that there was no evidence of an effect?

I was not sure what the authors were trying to convey with the last bullet point under “Strengths & limitations”.

Introduction

L71: please provide citations to support this statement, specifically regarding corroboration of findings by prospective studies.

L77-78: revise “an individuals”.

L78: a possible issue with this comparison is that the causal variant is not known because results from GWAS that are used for MR do not provide definitive evidence of what the causal variant is, instead they identify regions that may contain causal variant(s).

L95: the recent study by Giannelis et al. 2021, J Affective Disorders, doi: 10.1016/j.jad.2020.10.017 used MR to examine the relationship between the number of children, cohabitation and depression in the UK Biobank. This study should be discussed here and will impact the authors’ framing of the current work as proof-of-concept study for studying such traits using MR. Well-being is of course correlated with mental health conditions such as depression, including in the UK Biobank (Fabbri et al., 2022, J Affective Disorders, doi: 10.1016/j.jad.2022.07.023; Fabbri et al., 2021, Psych Medicine, doi: 10.1017/S003329172100502X).

L107-109: please clarify what the authors mean by “active gene-environmental correlation within an MR study design”.

Methods

It would be useful to distinguish between biological and adopted children if this difference was considered in the underlying studies.

L130: “fully anonymised” should be revised to “pseudo anonymised”. Given the availability of whole genome sequencing data, the statement that “data from the UKB are fully anonymised” is not also correct in my view. Perhaps this sentence could be removed?

A lot of methodological details are provided elsewhere which makes the current work difficult to read as a stand-alone paper.

Please specify in the methods section (not only in Figure 3) how SNPs were selected for the main analysis, i.e., the p-value threshold. This will also help readers better understand section 2.5.3.

Various sections could be revised for clarity. I also found that more guidance could be provided to readers who may not be familiar with all the methodological details and jargon, e.g., L194-196, the section on negative controls or L248 (the wording will only make sense to readers familiar with DAGs).

Does the current report adhere to CONSORT-MR guidelines?

L235: revise “latter” to “later”.

More details could be provided in section 2.5.4.

Results

L273: revise “sage” to “stage”.

I am concerned about the large sample overlap and how this might bias the results reported.

Regarding the wording in section 3.2, please see my comment above regarding the abstract.

L331: revise “is” to “was”.

The authors report a lot of interpretation of their findings in the results section (e.g., section 3.5), which might be more appropriate for the discussion.

L360-362: these estimates relate, so the argument may be somewhat circular?

Discussion

When talking about a “genetic liability to having children”, what would be possible mechanisms mediating such effects?

L385: the average age of the UK Biobank participants is not around 60, but closer to mid 50s. This could be due to reporting the mean age at the assessment of wellbeing. Please clarify.

L407-409: see Giannelis et al paper cited above.

L418 (following): I was not fully convinced by the argument that good quality individual data were not available. For example, the UK Biobank has multiple data fields that can be used to derive estimates of well-being.

L430: add “we”.

Reviewer 2 Report

What a pleased to read this MR analysis on children on wellbeing. The design and analyses are well thought through and conducted. Below a few minor suggestions.

Page 2, line 70: Can you provide more details on strengths/weaknesses of the available prospective studies.

I was stunned that the authors chose ‘number of siblings’ as a negative control. After reading the justification and thinking about DAG 2, I think this is a brilliant way to examine dynastic effects.

Main analysis using 6 SNPs: MR PRESSO and RAPS are suitable 10+ SNPs (e.g. 10.1002/gepi.22295). Consider using PRESSO & RAPS for the 5x10-6 analyses with 10+ SNPs.

Page 8 line 302, please replace I² with I²gx. Otherwise the reader might confuse Bowdens I²gx with Higgens I². Describe I²gx in the stats section.

Page 11, line 430: “Finally, …”. Add, “we”.
